# Magnetic Field Effect in Bimolecular Rate Constant of Radical Recombination

**DOI:** 10.3390/ijms24087555

**Published:** 2023-04-20

**Authors:** Alexander B. Doktorov, Nikita N. Lukzen

**Affiliations:** 1International Tomography Center SB RAS, 630090 Novosibirsk, Russia; doktorov@kinetics.nsc.ru; 2Voevodsky Institute of Chemical Kinetics and Combustion SB RAS, 630090 Novosibirsk, Russia; 3Physics Faculty, Novosibirsk State University, 630090 Novosibirsk, Russia

**Keywords:** radicals, spin-selective radical recombination, magnetic and spin effects in chemical reactions, diffusion-influenced reactions, spin chemistry

## Abstract

The influence of magnetic fields on chemical reactions, including biological ones, has been and still is a topical subject in the field of scientific research. Experimentally discovered and theoretically substantiated magnetic and spin effects in chemical radical reactions form the basis of research in the field of spin chemistry. In the present work, the effect of a magnetic field on the rate constant of the bimolecular spin-selective recombination of radicals in the bulk of a solution is considered theoretically for the first time, taking into account the hyperfine interaction of radical spins with their magnetic nuclei. In addition, the paramagnetic relaxation of unpaired spins of the radicals and the non-equality of their g-factors that also influence the recombination process are taken into account. It is found that the reaction rate constant can vary in magnetic field from a few to half a dozen percent, depending on the relative diffusion coefficient of radicals, which is determined by the solution viscosity. It is shown that the consideration of hyperfine interactions gives rise to the presence of resonances in the dependence of the rate constant on the magnetic field. The magnitudes of the magnetic fields of these resonances are determined by the hyperfine coupling constants and difference in the g-factors of the recombining radicals. Analytical expressions for the reaction rate constant of the bulk recombination for magnetic fields larger than hfi (hyperfine interaction) constants are obtained. In general, it is shown for the first time that accounting for hyperfine interactions of radical spins with magnetic nuclei significantly affects the dependence of the reaction rate constant of the bulk radical recombination on the magnetic field.

## 1. Introduction

The influence of constant and variable magnetic fields on chemical reactions continues to draw the attention of researchers. This is related both to general scientific interest and the fact that modern humans are constantly surrounded by these fields. Hence, question that are, for instance, interested in the connection with the safety of magnetic field influence in the case of MRI diagnostics can arise. So far, the only scientifically justified effect of a magnetic field on chemical reactions is the effect a magnetic field has on radical recombination reactions [1,2,3,4]. The Zeeman energy of the radical spin is too small even in magnetic fields of the order of tens of Tesla, and cannot significantly affect the thermodynamics of radical reactions. However, the magnetic field can and does affect the kinetics of the reactions and the rate of reactions, even with the formation of a temporary thermal non-equilibrium concentration of reactants. The influence of the magnetic field on the rate of reactions is important in many aspects, such as in the scientific-fundamental, technological, and biological aspects. Thus, from a general point of view, the life activity process of a living organism is determined by many reactions, including those involving free radicals; therefore, the influence of a magnetic field on their rate is very important. An example of this is the process of lipid peroxidation, where there are stages of recombination of lipid radicals [5].

Theoretical and experimental studies of magnetic field effects, constant and variable, on chemical radical reactions, including photochemical and radiation-chemical reactions, are the subject of spin chemistry [1,3,4,6,7,8]. Spin chemistry events are based on the interplay of three phenomena. The first phenomenon is the spin selectivity of the radical recombination reaction [9]. This phenomenon exists because the total spin of a pair of radicals is conserved during their recombination. Since radicals recombine, as a rule, into a diamagnetic product, this means that they can recombine only from the collective singlet spin state, that is, when the spins of the radicals are antiparallel. Recombination from the triplet state can happen as well, but at a different rate than for the singlet state, i.e., in this case, there is also a spin-selective nature of recombination. The second phenomenon is the encounter of reactants (radicals) in solution consisting of many re-contacts [10]. In contrast to the gas phase, where the encounter of reactants consists of a sole single collision, while in solution, the radical encounter consists of many re-contacts. The reactants diffusively separate from each other after another contact, but it is highly probable that the next re-contact follows. This mechanism of reactant encounter in solution is determined by where the reactants are situated in a “cage” formed by the solvent molecules. The third factor determining the influence of the magnetic field on radical reactions is the evolution of the spins of each of the radicals in the presence of external magnetic fields due to the magnetic interactions of electronic spins with the magnetic nuclei in each of the radicals. If, for instance, radicals have different g-factors, then the joint (collective) spin system of the radical pair (RP) experiences single–triplet transitions; the frequency of these transitions is proportional to the external magnetic field. Thus, if a RP first meet in solution in the non-reactive triplet state, it does not mean that the radicals of RP can escape into a bulk of the solution without recombination. In the time interval between the first and second contacts, and generally between subsequent re-contacts, the pair of radicals, due to the transitions between triplet and singlet states induced by the magnetic field and magnetic interactions, can converse the reactive singlet state and recombine. Thus, the recombination probability at a radical’s encounter depends on the frequency of transitions between singlet and triplet states, which is determined, among other things, by the external magnetic field, which determines the dependence of recombination efficiency and, consequently, the radical recombination rate on the magnetic field. The recombination probability in the course of the encounter is also determined by the re-contacts statistics and thus by the combination (interplay) of all three announced phenomena, in which the external magnetic field is a kind of moderator. 

The described mechanism of interplay of these three phenomena, which explains the influence of the magnetic field on the recombination of radicals, is called the radical-pair mechanism, which was first independently proposed by Closs and by Kaptein and Oosterhoff [11,12,13]. This mechanism is detailed in the spin chemistry literature [1,2,3,6,7,8,14,15,16]. Experimentally, the influence of the magnetic field on radical reactions has been confirmed in many works, see [1,6,17,18,19,20,21,22,23,24,25,26,27,28,29]. In particular, magnetic effects were found in the oxidation of hydrocarbons in solution within reactions with stages of radical recombination similar to the radical stages of lipid peroxidation [20,21]. It has been suggested that the avian compass mechanism relies on magneto-sensitive radical pairs formed by photo-induced intramolecular electron transfer reactions in an array of aligned photoreceptors located in the retina [16,30]. There is also an indication of the magnetic field influence on the rate of DNA synthesis via the radical-pair mechanism [31].

In photo- and radiation-induced radical recombination reactions, a distinction is made between geminate and bulk bimolecular reactions. Geminate radical recombination in spin chemistry refers to the formation and possible recombination of a pair of radicals formed during the cleavage, for example, of a parent molecule, which, in a certain spin state, forms two radicals in the contact in the same spin state as the parent molecule. However, in the case of bulk recombination, it is assumed that the radicals are uniformly distributed in the bulk of the solution at the initial moment of time and that their spin state is uncorrelated upon contact of any radical pairs. After the geminate stage, the bulk stage of the recombination reaction follows, i.e., when the recombination of radicals that have not reacted in geminate pairs have escaped into the solvent, volume occurs. Obviously, these radicals have already met each other in an uncorrelated spin state.

In recent years, significant progress has been made in experimental techniques to observe radical intermediates of reactions over the time range from a few nanoseconds to microseconds [32,33,34], with high-resolution time. This enables the experimental observation of magnetic effects in the bulk recombination stage of radicals, i.e., radicals that have escaped into the bulk from geminate pairs.

In this work, we theoretically study the effect of the magnetic field and magnetic interactions of the unpaired electron spins of radicals with the external magnetic field and magnetic nuclei, as well as the effect of paramagnetic relaxation of electron spins on the bulk recombination of these radicals. Only a few works detailing the magnetic field effect on the bulk bimolecular recombination of radicals are available [35,36,37,38,39,40,41,42,43,44], in contrast to the large number of theoretical works on the magnetic effects in geminate recombination [45,46,47,48,49,50,51,52,53]. The theoretical study of the magnetic effect, i.e., the effect of the magnetic field on the bulk recombination constant of radicals, was first performed in [35,36]. In these works, it was assumed that radicals move by means of diffusion. The singlet–triplet transitions in a pair of the radicals encountered in solution occur due to the differences in Larmor frequencies in radical spins in the magnetic field, which occur when the g-factors of radicals are unequal. The same mechanism of singlet–triplet transitions was considered for bulk recombination in later works [44]. This mechanism of singlet–triplet mixing and the magnetic effect caused by it is called the Δg mechanism. In further studies [37,43], along with the Δg mechanism, the paramagnetic relaxation of the spins of one of the two radicals, which also significantly affects the singlet–triplet transitions and, as a result, the magnetic effect in the rate constant of the bulk recombination reaction, was also taken into account. This result was very useful for the interpretation of the magnetic field effect in the bulk recombination of the NO and $O_{2}^{-}$ radicals [54], where *g*-factor difference is on the order of unity, and the *T*_1_ relaxation time of the *NO* radical is very short and constitutes the order of one picosecond. Such a short spin relaxation should give rise to the disappearance of the magnetic effect. Nevertheless, a noticeable magnetic effect is observed because there is a large difference in the *g*-factors, which leads to a large difference in the Larmor frequencies in magnetic fields of the order of several Tesla, and this competes with relaxation. A theoretical study which considered the Δg mechanism and paramagnetic relaxation [43] was also applied in the same work to interpret the dependence of the rate constant of the bimolecular recombination of the complex of ruthenium radicals with bipyridine $Ru{\left(bpy\right)}_{3}^{3+}$ and methyl viologen $MV^{+}$ on viscosity and magnetic field. The relaxation time of paramagnetic ruthenium complexes was several dozen picoseconds, and the *g*-factor difference was also approximately equal. In Refs. [38,41,42], the magnetic effects in the recombination of radicals diffusing on the plane were considered (i.e., the two-dimensional case of radical recombination). In particular, in our work [42], the influence of paramagnetic relaxation on recombination, hyperfine interactions of radical spins with magnetic nuclei, and the difference in the g-factors of radicals were taken into account as well. This model addresses the possible magnetic effects in the recombination of lipid radicals that producer a diffusion motion (lateral diffusion) across the cell membrane surface. 

In this work, we have, for the first time, theoretically considered the magnetic field effect on the rate constant of the bulk recombination of radicals in arbitrary magnetic fields, taking into account the hyperfine couplings of radical spins with magnetic nuclei and the paramagnetic relaxation of radical spins. Accurately accounting for hyperfine couplings is especially important in relatively weak magnetic fields of the order of 1 mT, when the difference in the Larmor frequencies of radicals, attributable to the difference of g-factors, is insignificant. Although, the hyperfine interactions of radicals with magnetic nuclei in a dipole–dipole interaction of radical spins and paramagnetic relaxation were taken into account earlier in Refs. [41,42], as they were related to the two-dimensional diffusion of radicals. The three-dimensional case of a diffusion-controlled bimolecular reaction differs significantly from the two-dimensional case. As Razi Naqvi [55] first discovered, in the two-dimensional case, there is no stationary (time-independent) rate constant, or, more precisely, it is equal to zero. This also applies to the case of the spin-selective recombination of radicals [42]. In addition, for the three-dimensional diffusion of radicals, the average value of the dipole–dipole interaction of radical spins is zero, which is not true for the two-dimensional case [41,42]. 

## 2. Results

### 2.1. Theory

The theory of diffusion-controlled reactions in solution began with the famous work of M. Smoluchowski [56], who showed that the bimolecular reaction rate constant $k_{D}$ of reactants of two sorts, *A* and *B*, freely diffuse in solution and react instantaneously if their contact is equal to:(1)$$k_{D}=4\pi RD$$
where R is the sum of the van der Waals reactant’s radii and D is the relative diffusion coefficient equal to the sum of the diffusion coefficients of particles *A* and *B*. Thus, the recombination of reactants is described by the equation:(2)$$\frac{dc_{A}(t)}{dt}=-k_{D}{\cdot c}_{A}\left(t\right)\cdot c_{B}\left(t\right)$$
where $c_{A}(t)$ and $c_{B}(t)$ are the concentrations of reactants *A* and *B* at the instant of time *t*. The diffusion constant $k_{D}$ is actually the rate constant with which reactants meet in solution (encounter rate constant), even if they do not react. As Collins and Kimball [57] later showed, when the reactants react with a probability which is less than the unity upon contact, the reaction rate constant k is given by the following expression:(3)$$k=\frac{k_{r}k_{D}}{k_{r}+k_{D}},$$
where $k_{r}$ is the so-called reaction constant. It can be seen from Equation (3), that the reaction rate constant is equal to the product of the diffusion constant $k_{D}$ and ratio $k_{r}/(k_{D}+k_{r})$, the latter is the reaction probability during all re-contacts within the single encounter of reactants. 

If the thickness ∆ of the layer near the contact where the reaction occurs is small and the reaction rate in the layer is $w_{0}$, then the reaction constant is equal to:(4)$$k_{r}=4\pi R^{2}{\cdot w}_{0}\cdot ∆$$

Returning to the problem of radical recombination, one can note that, even without taking into account singlet–triplet transitions and only taking into account the fact that the recombination of radicals encountered in solution occurs only from their collective singlet state, the rate constant in Equation (3) should be multiplied by the ¼ factor that is the so-called spin factor:(5)$$k=\frac{1}{4}\frac{k_{r}k_{D}}{k_{r}+k_{D}}$$

This factor is statistical, i.e., it reflects the statistics of the spin states of a pair of radicals (or a radical pair (RP)). This is because a radical pair has four spin states, three of which are triplets and one is a singlet, and they are equally probable (if we do not consider the Boltzmann factor in the population of spin states, which is correct for fields smaller than several Tesla). Thus, the ¼ factor in Equation (5) is the probability that the radicals encounter in the singlet, reactive state. If one neglects paramagnetic relaxation and assumes that singlet–triplet transitions are due only to *g*-factor differences in a strong magnetic field, Equation (5) is modified as:(6)$$k=\frac{1}{2}\cdot \frac{\frac{k_{S}}{2}k_{D}}{\frac{k_{S}}{2}+k_{D}}$$

Here, similarly to Equation (4), $k_{S}=4\pi R^{2}{\cdot w}_{S}\cdot ∆$—is the reaction rate constant of recombination from the singlet state, where $w_{S}$ is the probability of recombination from the singlet state per unit time (recombination rate). We assume that the recombination occurs only from the singlet state, R и ∆ as in Equation (4), are the contact radius and reaction layer thickness, respectively. The result (6) is obtained from Equation (40) of Ref. [43], when the radicals have different values of *g*-factors and for the case of a strong magnetic field so that: (7)$$\left|\Delta g\right|\cdot \beta_{e.}\cdot B\cdot \tau_{D}\gg 1$$

Here, Δg is the difference of radicals’ *g*-factors, $\beta_{e}$ is Bohr magneton, B is magnetic field and $\tau_{D}=\frac{R^{2}}{D}$ is a complete time of all re-contacts, i.e., encounter time equal to the residence time of the radical pair in the “cage” of the solution.

In the derivation of Equation (6), the hyperfine couplings of the electron spins with the spins of magnetic nuclei were not taken into account. The difference in the Larmor frequencies of the radicals gives rise to the transition between only the singlet state *S* and the triplet state $T_{0}$, with zero spin projection onto the direction of the external magnetic field. Therefore, Equation (6) can be interpreted as follows: The factor 1/2 in front of the right hand side expression is the statistical weight of the event that the radicals will meet either in the singlet state *S* or in the triplet state $T_{0}$. In turn, the halving of the reaction constant $\frac{k_{S}}{2}$ can be rationalized as the fact that, if the radical pair already met in the *S* or $T_{0}$ states, then, due to fast transitions between these states, their populations at any instant of time are equal, but since the reaction occurs only from the singlet state the $k_{S}$ constant effectively decreases by half.

In this paper, we theoretically study the bulk recombination of two types of radicals in solution, each containing one magnetic nucleus. We also assume that the radicals have different values of g-factors. We also consider paramagnetic relaxation with longitudinal relaxation times $T_{1A},T_{1B}$, and transverse times $T_{2A},T_{2B}$, with index *A* referring to the first type of radical and index *B* to the second type of radical.

To calculate the recombination rate constant, we used the so-called Encounter theory [35,37,43,45,58,59,60,61,62,63,64,65,66,67,68,69,70,71], which is a generalization of the theory of diffusion-influenced reactions in the liquid phase for reactions of reactants with an internal quantum, particularly spin structure. In this theory, the equations for the ${\hat{\sigma }}_{A}(t)$ and ${\hat{\sigma }}_{B}(t)$ density matrices, which are normalized to radical concentrations, have the following general form:(8)$$\begin{array}{c} \frac{d{\hat{\sigma }}_{A}(t)}{dt}={\hat{\hat{L}}}_{A}\sigma_{A}\left(t\right)+Tr_{B}\left({\hat{\hat{K}}}_{AB}(t)\sigma_{A}\left(t\right)⊗\sigma_{B}\left(t\right)\right),\\ \frac{d{\hat{\sigma }}_{B}(t)}{dt}={\hat{\hat{L}}}_{B}\sigma_{B}\left(t\right)+Tr_{A}\left({\hat{\hat{K}}}_{AB}(t)\sigma_{A}\left(t\right)⊗\sigma_{B}\left(t\right)\right) \end{array}$$

Here, ${\hat{\hat{L}}}_{A}$ and ${\hat{\hat{L}}}_{B}$ are the Liouvillian operators, governing the internal evolution of the spin system for each of the radicals A and B, including the Zeeman interaction of electron spins with an external magnetic field, hyperfine interactions of the electron spin and spins of the magnetic nuclei, and paramagnetic relaxation. The traces of the one-particle density matrices at time $t,\sigma_{A}\left(t\right)$ и $\sigma_{B}\left(t\right)$ are the concentrations $C_{A}\left(t\right)$ and $C_{B}\left(t\right)$ of radicals *A* and *B*, respectively, that is:(9)$$Tr\left(\sigma_{A}\left(t\right)\right)=C_{A}\left(t\right);Tr\left(\sigma_{B}\left(t\right)\right)=C_{B}\left(t\right)$$

In Equation (8) ${\hat{\hat{K}}}_{AB}(t)$ is a bimolecular operator, which is calculated through the evolution equations of the RP density matrix during the encounter. The sign ⊗ denotes the Kroneckerian or tensor product of the matrices. The procedure for calculating ${\hat{\hat{K}}}_{AB}(t)$, which takes into account diffusional relative motion of radicals (or some other type of stochastic motion of radicals), interaction of radical spins with a magnetic field, paramagnetic spin relaxation, and the recombination of radicals from the singlet state, are described in detail in Ref. [42].

We calculated the operator ${\hat{\hat{K}}}_{AB}(t)$ in the contact approximation [64,72], i.e., we considered that the recombination reaction from the singlet state occurs in a narrow layer ∆≪*R* (see Equation (4)). We are interested, of course, in the equations for radical concentrations rather than in the detailed kinetics of all of the elements and their one-particle density matrices. It turned out that, in the case of radical recombination that we are considering, Equation (8) can be “closed” to only include radical concentrations:(10)$$\frac{dC_{A}(t)}{dt}=-k\cdot C_{A}\left(t\right)\cdot C_{B}\left(t\right),\frac{dC_{B}\left(t\right)}{dt}=-k\cdot C_{B}\left(t\right)\cdot C_{B}\left(t\right).$$

In this paper, we study the dependence of the recombination rate constant k on the magnetic field at various values of the relative diffusion coefficient, paramagnetic relaxation times, and hyperfine couplings in radicals, assuming that each radical contains only one magnetic nucleus.

Thus, the Hamiltonian $\hat{H}$, of the spin system of RP is:(11)$$\hat{H}=g_{A}\beta_{e}B{\hat{S}}_{Az}+g_{B}\beta_{e}B{\hat{S}}_{Bz}+a_{1}{\vec{\hat{S}}}_{A}{\vec{\hat{I}}}_{A}+a_{2}{\vec{\hat{S}}}_{B}{\vec{\hat{I}}}_{B},$$
where the indices A and B refer to the radical A and B, respectively, $g_{A},g_{B}$ are g—factors of radicals, ${\vec{\hat{S}}}_{A}$, ${\hat{S}}_{Az}$ и ${\vec{\hat{S}}}_{B}$, ${\hat{S}}_{Bz}$—electron spins operators and operators of their projections on the z axis, which is aligned along the magnetic field B, ${\vec{\hat{I}}}_{A}$, ${\vec{\hat{I}}}_{B}$- spin operators (spin of each nucleus equal to ½) of magnetic nuclei, $a_{1},a_{2}$-hyperfine coupling constants of electron spins of radicals with the magnetic nuclei. In Equation (11) we neglect the Zeeman interaction of nuclei spins with the external magnetic field since they are negligible compared to other interactions. In a high magnetic field, when $B\gg a_{1},a_{2}$, one can neglect by terms $a_{1}I_{Ax}S_{Ax},a_{1}I_{Ay}S_{Ay}$ and $a_{2}I_{By}S_{By},a_{2}I_{By}S_{By}$ in scalar products $a_{1}{\vec{\hat{S}}}_{A}{\vec{\hat{I}}}_{A}$ и $a_{2}{\vec{\hat{S}}}_{B}{\vec{\hat{I}}}_{B}$, since they are responsible for the simultaneous flip-flop of electron and nuclear spins, i.e., they transition to higher- or lower-lying energy states by a value almost equal to the Zeeman energy of the electron spin. In this high-field approximation, the Hamiltonian is reduced as follows:(12)$$\hat{H}=g_{A}\beta_{e}B{\hat{S}}_{Az}+g_{B}\beta_{e}B{\hat{S}}_{Bz}+a_{1}S_{Az}I_{Az}+a_{2}S_{Bz}I_{Bz}$$

In fact, Equation (12) means that each of the spins experience a total magnetic field equal to the sum of the external magnetic field and the hyperfine magnetic field of the nuclei, which is also directed along the *z*-axis. For the first spin, this field is $B+\frac{a_{1}I_{Az}}{g_{A}\beta_{e}}$, for the second one $B+\frac{a_{2}I_{Bz}}{g_{B}\beta_{e}}$, and the effective Larmor frequencies $\omega_{A}$ and $\omega_{B}$ of the spins are equal to:(13)$$\omega_{A}=g_{A}\beta_{e}B+a_{1}I_{Az},\omega_{B}=g_{B}\beta_{e}B+a_{2}I_{Bz}$$

Each value of the projections $I_{Az}$ and $I_{Bz}$ can take one of the two values: $I_{Az}=\pm \frac{1}{2},$ $I_{Bz}=\pm \frac{1}{2}$. It can be seen that the difference in these effective Larmor frequencies is not equal to zero in the high-field approximation, even for equal *g*-factors of the radicals. The difference in the Larmor frequencies causes, as mentioned above, transitions between the singlet state ***S*** and the triplet state $T_{0}$ (singlet–triplet mixing). In the high-field approximation (12), we obtained an analytical expression for the bimolecular radical recombination rate constant, also taking into account the paramagnetic relaxation of the radical spins (see Appendix A).

### 2.2. Magnetic Field Effects Calculation Results

Figure 1 shows the numerical calculation of the recombination rate constant dependence on the magnetic field, which was obtained via analytical equations (Equations (S2) and (S3) of the Appendix A). Each of the radicals is considered to have only one magnetic nucleus with spin ½. For the sake of simplicity, only the effect of relaxation on one of the radicals is taken into account; the difference in the g-factors of the radicals is equal to 0.001. One can see that the exact numerical calculation coincides with the analytical result obtained in the high-field approximation, when the magnetic field *B* > 20 mT (see Figure 1 and the right insertion in it). At the same time, at a lower magnetic field, and especially at magnetic field of zero, the exact value of the rate constant differs significantly from the value obtained in the high-field approximation if we formally set *B* = 0 for it. In view of this, the high-field approximation cannot be used to estimate the magnetic field effect in the reaction rate constant, i.e., to estimate the magnetic field dependence of *k(B)/k(B* = 0). However, the advantage of high-field approximation is that there is an analytical expression for it. The result of the high-field approximation can be used to compare the reaction rate constant for different magnetic fields, if the values of these fields fall within the applicability range of the high-field approximation, i.e., when the magnetic field is much larger than the hyperfine coupling constants in radicals. The left inset in Figure 1 shows that the limiting value of the rate constant, its “saturation,” is reached at extremely high fields, much higher than the maximum achievable value of the magnetic field 45.5 T [73], for which the experimental setup enabling such experiments has been realized. Figure 1 shows that the behavior of the rate constant on the magnitude of the magnetic field is not monotonous; instead, it has peculiarities, i.e., resonances. The fact is that, in a magnetic field that is much larger than the hyperfine coupling constants, singlet transitions occur mainly due (not counting paramagnetic relaxation) to the difference in the effective frequencies of electron spin precession in the external magnetic field and the hyperfine magnetic field of the nuclei, and the resonances are observed when this difference is equal to zero. This difference $\Delta \omega_{L}$ in precession frequencies is defined by Equation (13):(14)$$\begin{array}{ll} \Delta \omega_{L}=\omega_{A}-\omega_{B} & =g_{A}\beta_{e}B+a_{1}I_{Az}-g_{B}\beta_{e}B-a_{2}I_{Bz}\\ & =\left(g_{A}-g_{B}\right)\beta_{e}B+a_{1}I_{Az}-a_{2}I_{Bz} \end{array}$$Each value of the projections $I_{Az}$ and $I_{Bz}$ takes one of two possible values: $I_{Az}=\pm \frac{1}{2}$ and $I_{Bz}=\pm \frac{1}{2}$, and such values impact the magnetic field B and projections $I_{Az}$, and $I_{Bz}$. For the frequency difference $\Delta \omega_{L}=0$, i.e., the singlet–triplet transitions occur only due to paramagnetic relaxation. This is reflected in the resonance behavior of the reaction rate constant at these values of the magnetic field. Thus, it is easy to check that $\Delta \omega_{L}=0$ for a pair of values of projections of nuclear spins $I_{Az}=-\frac{1}{2}$ and $I_{Bz}=+\frac{1}{2}$ and B = 500 mT, and for the $I_{Az}=+\frac{1}{2}$ and $I_{Bz}=+\frac{1}{2}$ and B=1500 mT also. For two other possible combinations, $I_{Az}=+\frac{1}{2}$, $I_{Bz}=-\frac{1}{2}$ and $I_{Az}=-\frac{1}{2}$, $I_{Bz}=-\frac{1}{2}$, $\Delta \omega_{L}$ does not turn to zero at any value of the magnetic field *B*. In the general case of several magnetic nuclei in each of the radicals, the values of the magnetic fields *B*, at which resonances can be observed, are determined by the condition $\Delta \omega_{L}=0$, that is:(15)$$\Delta \omega_{L}=\omega_{A}-\omega_{B}=g_{A}\beta_{e}B-g_{B}\beta_{e}B+\sum_{i}a_{i}I_{iz}-\sum_{j}a_{j}I_{jz}=0$$

Here, index “*i*” represents the magnetic nuclei belonging to the first radical and index “*j*” represents the nuclei of the second radical, with $a_{i}$ being the hfi coupling constant of the *i*-th magnetic nucleus of the first radical and $I_{iz}$ being the z- projection of its spin; the similar notations are for the second radical.

Equation (15) shows that resonances exist only for a fraction of all of the possible combinations of nuclear spin projections. Nevertheless, this gives information (albeit incomplete) on the hyperfine structure of the recombining radicals. Additionally, we note that the difference between the lines of the EPR spectra of the radical is in the order of the values of the hfi coupling constant. At the same time, the differences between resonances in the influence of the magnetic field on the rate constant are in the order of the ratio of the hfi coupling constant to the difference in the g-factors of the radicals, i.e., the “spectrum” effect of the magnetic field has a higher spectral resolution.

Figure 2 shows the dependence of the magnetic field effect for the recombination rate constant, i.e., the ratio k(B)/k(B=0) on the magnetic field. It can be seen that the minimum value of this ratio is at the point of the first resonance *B* = 500 mT. The difference of the rate constant in this field from the rate constant in the zero magnetic field is about 3%, i.e., the maximum absolute value of the magnetic field effect is ≈3%. It is important to note that, at magnetic field values of the order of the hfi coupling constants, there is a narrow (by magnetic field sweep) feature—First, as the rate constant grows by a small value (the value of the magnetic effect is greater than unity) and then (before the first resonance) drops rapidly. This is the so-called low-field feature [14].

Figure 3 shows the dependence of the magnetic field effect for different diffusion coefficients, which are determined by the viscosity of the solution. It can be seen that, as the diffusion coefficient decreases, the amplitude of the magnetic effect increases (up to more than 15% for $D=2.0\cdot 10^{-8}\frac{cm^{2}}{s}$). Resonances in the 500 mT and 1500 mT fields are observed for all diffusion coefficients, increasing their depth as the diffusion coefficient decreases. The increase in the amplitude of the magnetic field effect in conjunction with the decreasing diffusion coefficient (increasing the viscosity of the solution) is due to the fact that the duration of the encounter of radicals in the solution is equal to $R^{2}/D$. Additionally, the time between re-contacts occurring during the encounter increases. This gives rise to a larger efficiency of singlet–triplet transitions, consequently engendering a larger magnetic effect.

The dependences of the magnetic field effect in the recombination constant for different values of the relaxation times $T_{1A}$ and $T_{2A}$ are shown in Figure 4; the diffusion coefficient is the same for all three dependences and is equal to $D=2.0\cdot 10^{-5}\frac{cm^{2}}{s}$. It can be seen that a shortening of the transverse relaxation time $T_{2}$ gives rise to a noticeable broadening of the resonances, while a shortening of the longitudinal relaxation time $T_{1}$ also gives rise to a decrease in their amplitude. At the same time, the amplitude of the magnetic effect, at least for the range of characteristic values of $T_{1}$ и $T_{2}$ times for radicals, changes insignificantly.

The dependences of the magnetic field effect for the case of equal values of hyperfine coupling constants $a_{1}=a_{2}=0.5$ mT in radicals for two different relative diffusion coefficients are shown in Figure 5$—D=2.0\cdot 10^{-5}\frac{cm^{2}}{s}$ (black line) and $D=2.0\cdot 10^{-8}\frac{cm^{2}}{s}$ (red line). It can be seen that there is only one resonance (not counting the low-field feature) in the magnetic field B=1000 mT, corresponding to the only combination of the nuclear spin projections of radicals $I_{Az}=+\frac{1}{2}$ и $I_{Bz}=+\frac{1}{2}$, at which $\Delta \omega_{L}=0$ in this field. It can be seen that at a larger diffusion coefficient, resonance exists, but it is rather weakly expressed.

Figure 6 plots the magnetic field effect in the recombination rate constant for radicals with the same *g* factors versus the magnetic field. The hfi coupling constants are equal to 0.5 mT for each of the radicals. Dependencies were calculated for two diffusion coefficients $D=2.0\cdot 10^{-5}\frac{cm^{2}}{s}$ (black line) and $D=2.0\cdot 10^{-8}\frac{cm^{2}}{s}$ (red line). No high-field resonances are observed in this case, as one would expect. However, there is a low-field feature, which is more pronounced at the diffusion coefficient $D=2.0\cdot 10^{-8}\frac{cm^{2}}{s}.$

The growth of the rate constant can be observed with the increase in the magnetic field compared to the value of the rate constant in the magnetic field of zero (magnetic effect is greater than unity). Subsequently, a decrease in the rate constant correlates with an increase in the magnetic field before reaching a plateau. The amplitude of the plateau is much larger at the diffusion coefficient $D=2.0\cdot 10^{-8}\frac{cm^{2}}{s}$.

## 3. Discussion

From dependences of the magnetic field effect (MFE) (i.e., the percentage change in the recombination rate constant in a magnetic field as compared to the recombination rate constant in a zero field) calculated above, it makes sense that its value varies from several to a half dozen percent in the range of magnetic fields from zero to 20 T. This range of magnetic fields is currently the most accessible for conducting such experiments. The maximum value of the magnetic effect essentially depends on the relative diffusion coefficient of the radicals, which is determined both by their size and by the solution viscosity. It is shown that accounting for the hyperfine interactions, which has not been done before, reveals the presence of resonances in the magnetic field effect when the *g*-factors of the recombining radicals are not the same. These resonances provide information about the hyperfine structure of the radicals, though this information is not complete. The $T_{1}$ relaxation time reduces the magnetic field effect because it causes a non-selective mixing of the populations of the collective singlet–triplet states of the radicals during their encounter in solution. Time $T_{2}$ has less effect on the magnitude of the magnetic field effect as its influence manifests itself in the broadening of the observed resonances in the magnetic field effect. We calculated the magnetic effects by assuming that the paramagnetic relaxation times are independent of the magnetic field. Strictly speaking, this is not true. The main relaxation mechanisms of electron spins for organic radicals are relaxation-induced via the stochastic modulation of the g-tensor anisotropy and the stochastic modulation of the anisotropic hyperfine interaction through rotational diffusion, which gives rise to the dependence of relaxation times, $T_{1}$ and $T_{2}$, on the magnetic field. However, as can be seen from our calculations, varying the relaxation times in a fairly wide range (from 250 ns to 1000 ns) does not significantly affect the magnitude of the magnetic effect; instead, it broadens it. The analytical equations obtained for the magnetic field effect in the high-field approximation coincide with the exact calculations in the magnetic fields larger than the hyperfine coupling constants of radicals. These high-field approximation expressions for the magnetic field effect cannot be used to calculate the magnetic effect, since they do not give the correct value of the rate constant when the magnetic field is zero. Nevertheless, they can be used to compare the recombination rate constant between different magnetic field magnitudes in the range of their applicability. In general, it is shown that taking into account the hyperfine couplings of the radical spins with magnetic nuclei have a significant effect on the dependence of the recombination reaction rate constant on the external magnetic field.

## 4. Materials and Methods

The numerical calculations were performed via the code written by authors in Matlab version 8.5 R2015a. The analytical calculation used the modern advanced method of Green’s function in the theory of physicochemical processes in liquid solutions.

## 5. Conclusions

The theoretical study of the magnetic field influence on the rate constant of spin-selective recombination of radicals in solution taking into account the hyperfine couplings of unpaired electrons with magnetic nuclei in radicals showed that the magnetic field effect on the recombination rate constant varies from several to half dozen percent, demonstrating the resonant behavior with respect to the magnetic field at different g-factor values of the radicals. The dependence of the radical recombination rate constant on the magnetic field, and the maximum achievable value of the magnetic field effect in the range of magnetic fields from zero to 20 T, significantly depend on the relative diffusion coefficient of recombining radicals, which is mainly determined by the viscosity of the solution. An analytical expression is obtained for the radical recombination rate constant for magnetic fields exceeding the hyperfine coupling constants. On the whole, it has been shown that taking into account hyperfine couplings is very important when calculating the radical recombination rate constant and its dependence on the magnetic field.

## Figures and Tables

**Figure 1 ijms-24-07555-f001:**
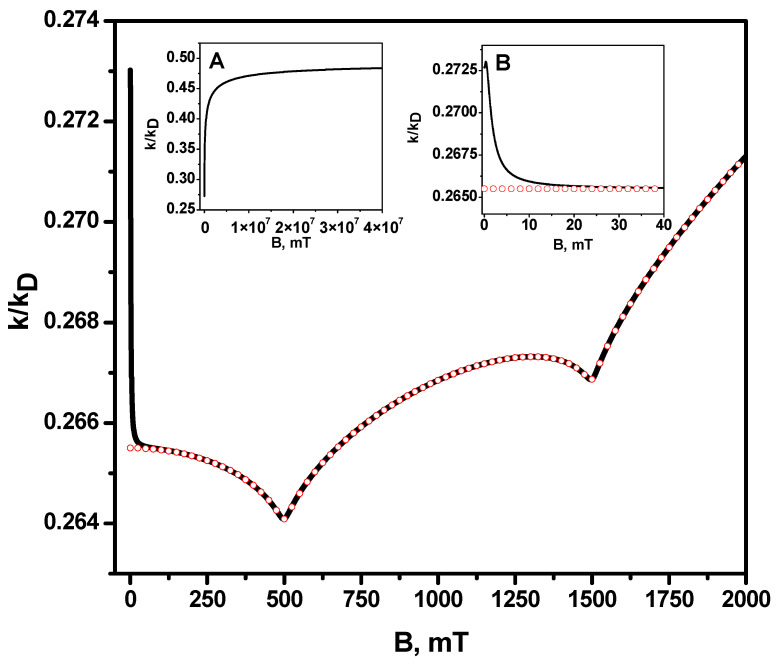
Dependence of the radical recombination rate constant in units of the diffusion-controlled reaction rate constant $k_{D}$ (diffusion constant) on the magnetic field. The solid black line is the exact numerical calculation, and the red open circles are the analytical calculation (according to Appendix A) in the high-field approximation. (**Insertion A**) is the rate constant for the range of very large magnetic fields; there, the high-field approximation totally coincides with the exact calculation. Therefore, dependencies are not distinguishable. (**Insertion B**) shows the recombination rate constant in a range of small magnetic fields. Model parameters: contact radius
R=10 Å, relative diffusion coefficient $D=2.0\cdot 10^{-5}cm^{2}/s$, hyperfine coupling constant in the first radical $a_{1}=0.5$ mT, hyperfine coupling constant in the second radical $a_{2}=1.0$ mT, g -factor of the first radical $g_{1}$ = 2.0, g-factor of the second radical $g_{2}$ = 2.001, $\frac{k_{S}}{k_{D}}=100$ (diffusion controlled regime). Longitudinal, $T_{1A}$, and transverse relaxation times $T_{2A}$ of the radical A are: $T_{1A}=1000$ ns, $T_{2A}=1000$ ns, we neglect by relaxation of B radical.

**Figure 2 ijms-24-07555-f002:**
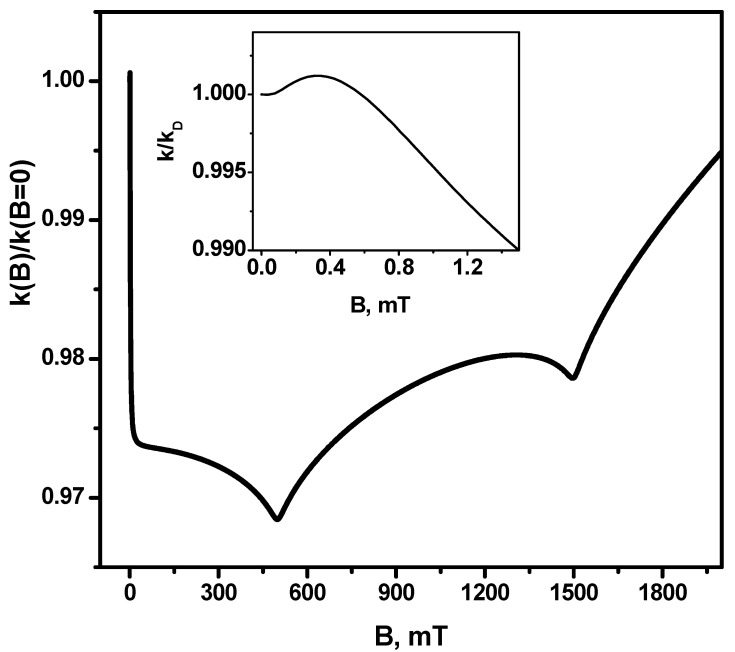
Magnetic field effect (MFE) in the recombination rate constant *k(B)/k(B* = 0). Insertion—magnetic field effect on the range of small magnetic fields. The other parameters are the same as in Figure 1.

**Figure 3 ijms-24-07555-f003:**
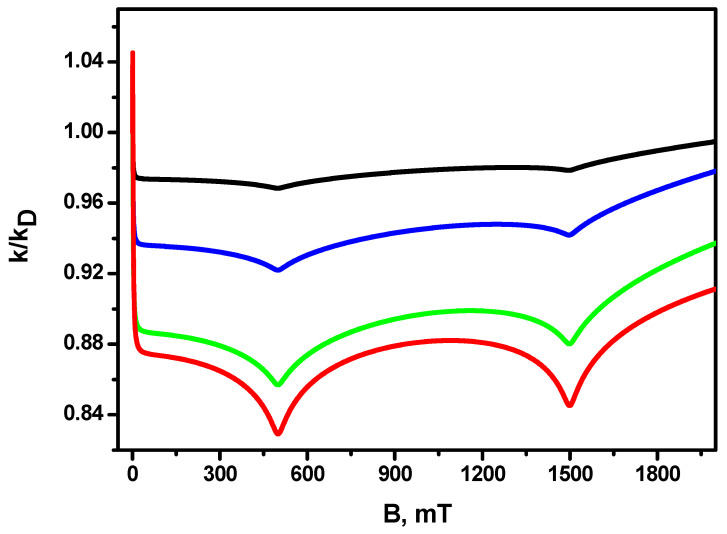
MFE in the recombination rate constant for different diffusion coefficients. Red line $D=2.0\cdot 10^{-8}\frac{cm^{2}}{s}$, green line $D=2.0\cdot 10^{-7}\frac{cm^{2}}{s}$, blue line $D=2.0\cdot 10^{-6}\frac{cm^{2}}{s}$, black line $D=2.0\cdot 10^{-5}\frac{cm^{2}}{s}$. The other parameters are the same as in Figure 1.

**Figure 4 ijms-24-07555-f004:**
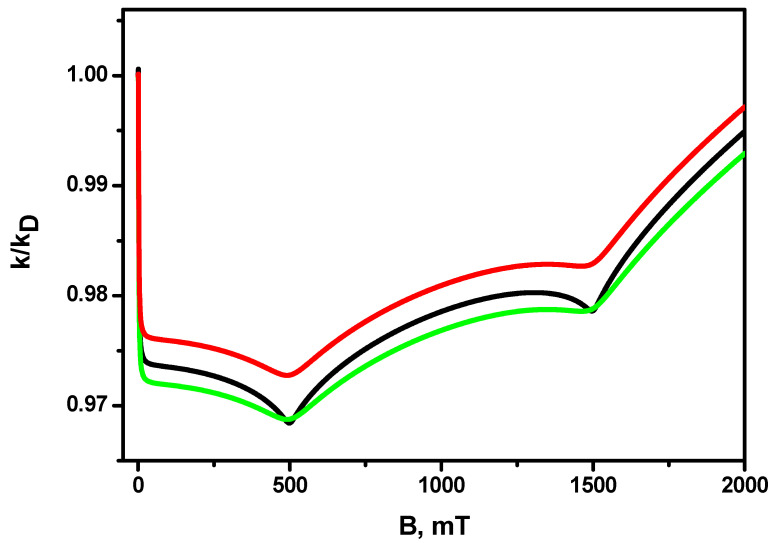
Magnetic field effect *k(B)/k(B* = 0) for different relaxation times. Black line: $T_{1A}=1000\;ns,T_{2A}=1000\;ns;$ green line: $T_{1A}=1000\;ns,T_{2A}=250\;ns$; red line: $T_{1A}=250\;ns,T_{2A}=250\;ns$. The other parameters are the same as in Figure 1.

**Figure 5 ijms-24-07555-f005:**
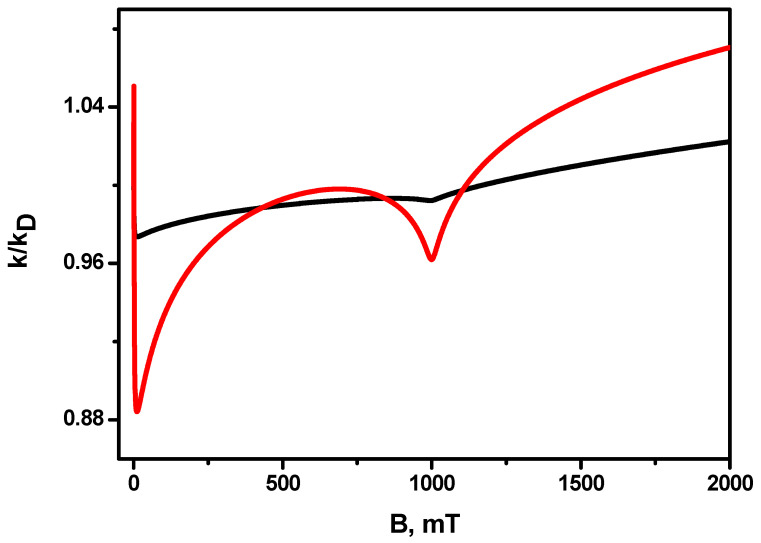
Magnetic field effect k(B)/k(B=0) for equal hyperfine constants $a_{1}=a_{2}=0.5\;mT$ in radicals. Black line $D=2.0\cdot 10^{-5}\frac{cm^{2}}{s}$, red line—$D=2.0\cdot 10^{-8}\frac{cm^{2}}{s}$. Other parameters are the same as in Figure 1.

**Figure 6 ijms-24-07555-f006:**
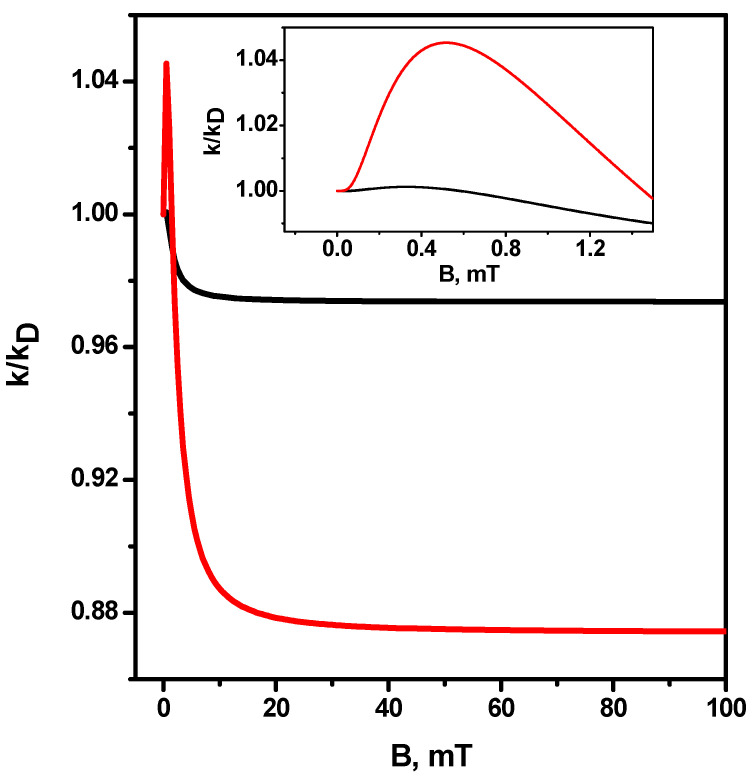
Magnetic field effect *k(B)/k(B* = 0) for equal g-factors of radicals, $g_{1}=g_{2}=2.0$ The black line corresponds to $D=2.0\cdot 10^{-5}\frac{cm^{2}}{s}$, the red line corresponds to $D=2.0\cdot 10^{-8}\frac{cm^{2}}{s}$. Other parameters are the same as in Figure 1. Insertion—the magnetic field effect on a smaller scale.

## Data Availability

Not applicable.

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
