# Peer review of "Magnetic Field Effect in Bimolecular Rate Constant of Radical Recombination"

_ijms, 2023, doi:10.3390/ijms24087555_

Round 1

Reviewer 1 Report

The manuscript titled Magnetic Field Effect in Bimolecular Rate Constant of Radical Recombination address the problem of the influence of magnetic fields on the rate constant of the bulk recombination of radicals. The work is based on computer modelling. Such kind of modelling is of growing interest as a little is known about impacts of magnetic fields on chemical reactions or even biological organisms.

                The Authors showed that the hyperfine couplings of the radical spins with magnetic nuclei have a significant effect on the dependence of the recombination reaction rate constant on the external magnetic field. Also they have shown that varying the relaxation times broadens magnetic effect but does not significantly affect the magnitude of it.

                The manuscript is well and clearly written. The figures are of good quality, however the decimal point of values should be given with dot i.e. instead e.g 0,274 should be 0.274.

I was also  a bit confused as there were no conclusions paragraph but discussion at the end.

                In overall I think that the Authors did a good job and the results are worth of publication.

Author Response

We are very grateful to the Referee for the useful comments on our manuscript and its evaluation.

Answers on the comments:

Comment 1: The manuscript is well and clearly written. The figures are of good quality, however the decimal point of values should be given with dot i.e. instead e.g 0,274 should be 0.274.

We agree with the comment. In the revised manuscript version all decimal point of values in the figures are now given with dot. Besides we change the scale step on Figure 5 from 1000 mT to 500 mT for better presentation.

Comment 2. I was also a bit confused as there were no conclusions paragraph but discussion at the end.

We have corrected this deficiency and added section “Conclusions” at the end of the manuscript. We agree with the Referee that this is important, especially for the convenience of a reader. 

Reviewer 2 Report

Doktorov and Lukzen have presented an informative theoretical study on the magnetic field effects on kinetics of radical recombination in solution. Their insights into the influences of hyperfine interactions is most interesting. The manuscript as is, is most appropriate for this IJMS special issue.

Author Response

We are very grateful to the Referee for evaluating our manuscript and for taking the trouble to review it.